# Pose-ICL: 3D-Aware In-Context Learning for Pose-Controllable Subject Customization

**Xuan Han** [* 1]  **Yihao Zhao** [* 1]  **Mingyu You** [1 2]

## Abstract

Subject Customization is a foundational task in modern image generation. By providing a few reference images and a text prompt, users can generate images of a specific object in any desired scene. However, existing methods still struggle to achieve effective pose control for customized subjects. In practice, they often exhibit inaccurate poses or inconsistent cross-pose appearances. These limitations suggest that understanding objects in a volumetric manner remains a significant challenge for 2D-native backbones. To address this challenge, we propose Pose-ICL, a tuning-free framework that leverages 3D-aware In-Context Learning (**ICL**) to directly adapt to new subjects through multiple paired image-pose references. Its core mechanism, Surface-Anchored Position Embedding (**SAPE**), equips the model with explicit 3D awareness by anchoring image tokens to the surface coordinates of a volumetric bounding box. Dedicated refinements ensure its seamless compatibility with existing DiT models. Extensive evaluations on both 3D assets and real-world subjects demonstrate that Pose-ICL significantly outperforms current methods in both pose accuracy and identity consistency.

## 1. Introduction

Synthesizing images of customized subjects is one of the most compelling applications of modern generative models (Rombach et al., 2022; Black Forest Labs, 2024; Chen et al., 2024). This technology serves a wide range of industries,

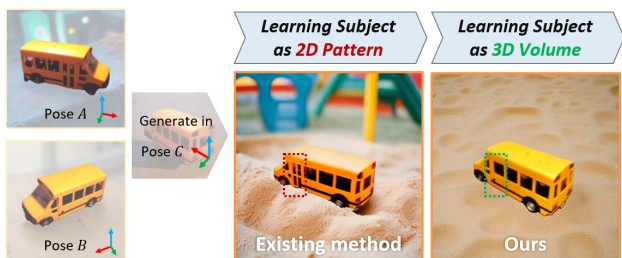

Figure 1. Conceptual comparison between existing methods and our 3D-aware approach. Unlike the methods that suffer from appearance inconsistency (**red box**), our approach maintains structural consistency and precise pose control through volumetric understanding (**green box**).

ranging from e-commerce to product design. By providing reference images of a specific object alongside a text prompt, users can generate images featuring the subject within any desired scene. In effect, customized image generation (Ruiz et al., 2023; Kumari et al., 2023; Li et al., 2023) liberates users from repetitive, low-level manual tasks, allowing for a greater focus on high-level creative expression.

Currently, advanced subject customization methods (Tan et al., 2025; Mi et al., 2025) can efficiently capture the semantics of target objects. However, how to achieve effective pose control for customized subjects remains an open challenge. Existing methods attempt to achieve pose control through internal 3D representations (Kumari et al., 2024) or auxiliary control branches (Qin et al., 2025). However, because these methods either struggle to bridge the gap between 3D geometry and 2D generation or treat customization as a decoupled task, they may encounter challenges in maintaining accurate poses and consistent appearance.

What, then, is the key to achieving effective pose control in subject customization? Fundamentally, changes in a subject's pose alter its appearance. When a generative model is tasked with modeling this relationship, it must recognize that the presence and placement of every visual detail are strictly governed by the laws of 3D perspective. Take the toy car in Figure 1 as an example: when generating Pose $C$ from reference Poses $A$ and $B$, the model must recognize that the door, now on the far side of the vehicle, should no longer be visible. Such a task necessitates modeling

*Equal contribution [1]College of Electronic and Information Engineering, Tongji University, Shanghai, China [2]State Key Laboratory of Autonomous Intelligent Unmanned Systems, Frontiers Science Center for Intelligent Autonomous Systems, Ministry of Education, Shanghai, China. Correspondence to: Mingyu You <myyou@tongji.edu.cn>.

*Proceedings of the 43rd International Conference on Machine Learning*, Seoul, South Korea. PMLR 306, 2026. Copyright 2026 by the author(s).

the subject as a 3D volume rather than a collection of 2D patterns. Existing methods, however, frequently lack such 3D awareness, which largely accounts for the previously noted inaccuracies in pose and appearance. As can be seen, they mistakenly render the far-side door on the near side. This analysis also reveals that the pursuit of pose control goes beyond expanding controllability, and is fundamentally essential for achieving more robust subject customization.

In this paper, we propose Pose-ICL to systematically address the above challenge. Our framework introduces a 3D-aware In-Context Learning (**ICL**) paradigm that achieves tuning-free, pose-controllable subject customization. Pose-ICL accepts multiple paired image-pose references of a customized subject, where poses can be obtained via automated estimation in practice. Tokens from all reference images are concatenated to serve as the input context for the model. Simultaneously, a volumetric bounding box is rendered for each pose. Its surface coordinates are transformed into Surface-Anchored Position Embeddings (**SAPE**), which are bound to the corresponding image tokens. This enhancement conveys rigid spatial constraints among local image features to the model, facilitating a volumetric understanding of the subject. Extensive evaluations are conducted on both 3D synthetic assets and real-world subjects. Pose-ICL achieves advanced performance relative to existing pose-control methods while offering enhanced cross-pose consistency compared to general customization baselines.

Our contributions can be summarized as follows:

- We provide a systematic analysis of the challenges in pose-controllable subject customization and pioneer Pose-ICL, a 3D-aware In-context Learning paradigm. To our knowledge, it represents the first tuning-free framework in this field.

- We introduce SAPE to equip the model with explicit 3D awareness. By embedding rigid spatial correspondences into position embeddings, it enhances the model's ability to achieve precise pose control while maintaining robust cross-pose consistency.

- We contribute a diverse dataset comprising both 3D assets and real-world subjects. A specialized generative pipeline is employed to enhance the visual quality and diversity of the images, providing a robust resource to facilitate future research.

## 2. Related Works

This section reviews recent developments in subject customization (Ruiz et al., 2023; Gal et al., 2023; Black Forest Labs et al., 2025; Tan et al., 2025) and pose-controllable generation (Kumari et al., 2024; Qin et al., 2025). Representative methods from these fields are included in our comparative experiments. To ensure a thorough comparison, editing-based pipelines (Wu et al., 2025; Xu et al., 2024) addressing similar tasks are also taken into consideration.

**Subject Customization** Pioneering works in this field, such as DreamBooth (Ruiz et al., 2023) and Textual Inversion (Gal et al., 2023), typically utilized multiple reference images of a target subject as training data. These methods bound the subject to a rare or modified word token through a tuning process. However, they often required hundreds of tuning steps and can be prone to reduced appearance fidelity due to inherent information loss. The introduction of DiT (Peebles & Xie, 2023) architectures enabled a paradigm shift toward in-context learning (Zhang et al., 2025). Methods such as OminiControl (Tan et al., 2025) treated reference images as context rather than training data. By establishing pathways that facilitated feature sharing between reference and target images. It successfully shifted the model's objective from memorizing the object to "copying" features from the context. Same paradigm was also adopted by advanced models such as FLUX-Kontext (Black Forest Labs et al., 2025). As previously noted, the primary limitation of existing methods is their tendency to treat subjects as 2D patterns rather than 3D volumes, which may leads to inconsistencies across diverse poses. These methods are included in our comparative experiments to validate the value of 3D-aware modeling for precise subject learning.

**Pose-controllable Image Generation** Methods in this field were broadly classified into two categories. The first category relied on internal 3D representations to achieve pose control. For instance, Custom Diffusion 360 (Kumari et al., 2024) maintained an embedded Feature NeRF (Mildenhall et al., 2021). It took parameterized poses as input and the resulting view-specific features were integrated into the generative model. The primary challenge for such methods lay in bridging 2D generative models and implicit 3D representations within a limited number of tuning steps. Insufficient optimization of either component potentially led to inaccurate pose control or artifacts in appearance. The second category adopted a more direct approach to control. A representative method was SceneDesigner (Qin et al., 2025). It also utilized the surface coordinates of 3D bounding boxes as pose control conditions. However, instead of employing positional embeddings, it injected these control signals into the model through a dedicated ControlNet (Zhang et al., 2023) branch. The differences between these two strategies will be discussed in the experimental section. Furthermore, because SceneDesigner treated pose control and subject customization as independent tasks, it required auxiliary methods like DreamBooth for customizing new objects. This work proposes that these two objectives can mutually reinforce each other within a more compact, tuning-free framework.

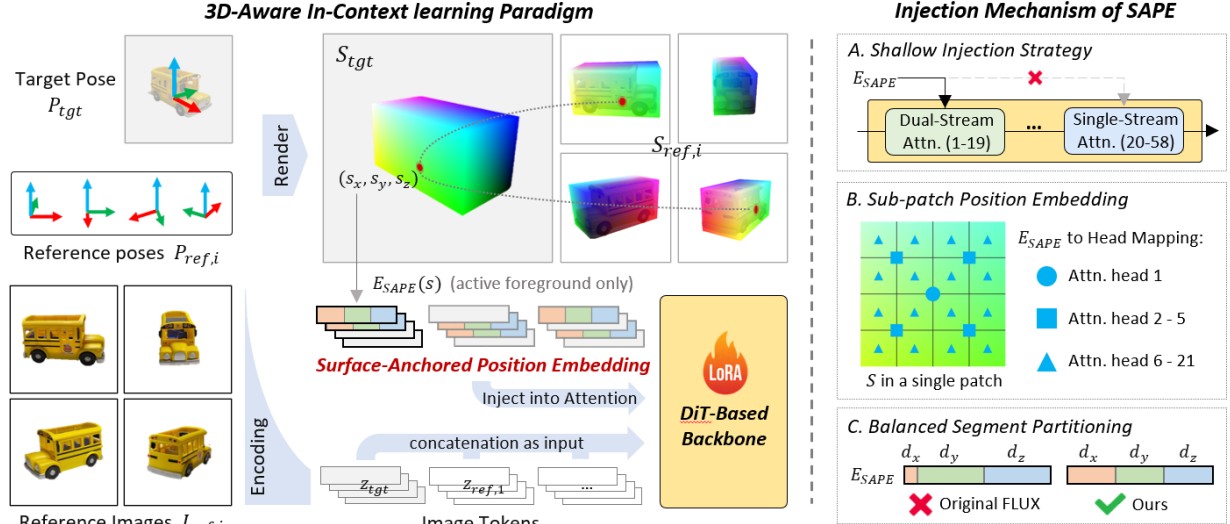

*Figure 2.* Overview of the Pose-ICL framework. (**Left**) Schematic of the 3D-aware in-context learning paradigm. (**Right**) The three refinements to the injection mechanism of SAPE. For clarity, SAPE is visualized as vectors and text prompts are omitted.

**Editing-based Pipelines** Multi-stage pipelines were another significant alternative for the task addressed in this work. This review primarily focused on two prevalent pipelines. The first involved utilizing large image-editing models (Han et al., 2025), such as Qwen-Edit (Wu et al., 2025), for sequential instruction-driven editing. Under this scheme, subjects were first rotated to specified poses via text prompts, followed by background completion. Due to limited 3D awareness, such methods were typically restricted to canonical views (e.g., front or side) rather than achieving continuous pose control. Another strategy utilized large reconstruction models (e.g., InstantMesh (Xu et al., 2024)) to generate 3D assets, which were then rendered and inpainted. In this pipeline, reference images contributed only to the reconstruction stage, precluding direct feature exchange with the generated image. Consequently, reconstruction inaccuracies might accumulate across subsequent steps, leading to a degradation in overall image quality.

## 3. Method

We propose Pose-ICL, a tuning-free framework designed for pose-controllable subject customization. The overall architecture is illustrated in Figure 2. The input to our task consists of a set of reference images $\{I_{ref,i}\}_{i=1}^{N}$ of the customized subject and the corresponding poses $\{P_{ref,i}\}_{i=1}^{N}$. In practice, the poses can be obtained via automated estimation methods. The objective is to synthesize a target image $I_{tgt}$ of the subject under a specified target pose $P_{tgt}$, where the scene semantics are conditioned on a text prompt. The proposed method is built upon the diffusion model, specifically adopting the DiT-based FLUX.1-dev (Black Forest Labs, 2024) as the backbone. To achieve tuning-free adapta-

tion to new subjects, a novel 3D-aware In-Context Learning (**ICL**) paradigm is employed. Within this paradigm, all reference images are also encoded into image tokens $\{z_{ref,i}\}_{i=1}^{N}$, which are fed into the DiT model as context alongside the target tokens $z_{tgt}$. The core mechanism of Pose-ICL for incorporating pose information is the Surface-Anchored Position Embedding (SAPE). It anchors the subject within a volumetric bounding box and transforms its surface coordinates maps $S$ into positional embeddings $E_{SAPE}(s)$. By providing explicit local correspondence cues, it facilitates a volumetric understanding of the subject across diverse poses. The remainder of this section details the formulation and injection of SAPE, followed by the training and sampling procedures of the 3D-aware ICL framework.

### 3.1. Surface-Anchored Position Embedding

The first step in formulating the SAPE is determining the volumetric bounding box of the customized subject. When estimating pose parameters using methods such as COLMAP (Schönberger & Frahm, 2016; Schönberger et al., 2016) or Fast3R (Yang et al., 2025), a 3D point cloud of the subject is simultaneously generated. Of note, these methods output camera poses, which are equivalent to subject poses assuming a static scene. The point cloud's oriented bounding box serves as the anchored volumetric box. In Pose-ICL, this box is axis-agnostic, meaning axes do not carry fixed semantic labels across different categories. The only requirement is a consistent box-to-object binding across all views of a specific subject. For instance, its $x$-axis may represent any primary dimension, such as the lateral or vertical span of the object within its local frame.

Subsequently, the box is rendered into surface coordinate

maps $S \in \mathbb{R}^{H \times W \times 3}$ according to the poses. As illustrated in Figure 2, the color of each pixel represents its normalized 3D surface coordinates $\mathbf{s} = (s_x, s_y, s_z) \in [0,1]^3$. These coordinates hold significant potential as indices for local features of subject. Integrating them into the subject learning process enables the model to better understand local feature correspondences across different poses.

In Pose-ICL, position embeddings serve as the vehicle for this input. Positional embeddings are fundamentally designed to inform transformer-based models of the spatial arrangement of tokens. Building on this, SAPE $E_{SAPE}$ leverages the Rotary Positional Embedding (**RoPE**) (Su et al., 2024) formulation. It is defined as a block-diagonal rotation matrix:

$$E_{SAPE}(\mathbf{s}) = \mathrm{diag}\left[R(s_x, d_x), R(s_y, d_y), R(s_z, d_z)\right] \quad (1)$$

$d = 128$ is aligned with the feature dimension of the backbone. As formulated, the SAPE matrix is partitioned into three segments $d_x, d_y, d_z$ ($d_x + d_y + d_z = d$), each corresponding to one of the spatial dimensions of the volumetric box. Each segment $R(s_i, d_i)$ is expanded as:

$$R(s_i, d_i) = \mathrm{diag}\left[\mathbf{M}_{i,0}, \mathbf{M}_{i,1}, \ldots, \mathbf{M}_{i, \frac{d_i}{2} - 1}\right] \quad (2)$$

The $2 \times 2$ rotation unit $\mathbf{M}_{i,j}$ is defined as:

$$\mathbf{M}_{i,j} = \begin{pmatrix} \cos(\gamma s_i \theta_j) & -\sin(\gamma s_i \theta_j) \\ \sin(\gamma s_i \theta_j) & \cos(\gamma s_i \theta_j) \end{pmatrix} \quad (3)$$

Where $\theta_j = 10000^{-2j/d_i}$ denotes the angular frequencies defined by the original RoPE. To ensure the frequency distribution aligns with the pre-trained DiT backbone, a scaling factor $\gamma = 32$ is introduced to modulate the range.

Consistent with other position embedding techniques in transformers, SAPE operates by being applied to the queries $q$ and keys $k$ within the attention layers, thereby modulating the attention scores $\alpha$ between them. The strength of modulation is dictated by the spatial distance between tokens. We adopt the RoPE formulation specifically because its modulating effect depends solely on the relative distance between tokens rather than their absolute positions. The mathematical process is formulated as:

$$\alpha = \mathrm{Softmax}\left( \frac{q^\top E_{SAPE}(\mathbf{s}_q)^\top E_{SAPE}(\mathbf{s}_k) k}{\sqrt{d}} \right) \quad (4)$$

The attention score naturally emphasizes tokens with close spatial region on the subject's surface, while suppressing interactions between distant regions, even if they exhibit similar 2D visual patterns. Furthermore, by using position

embeddings as the input vehicle, the 3D coordinates influence features indirectly through attention modulation. This prevents non-image signals from directly interfering with the subject's visual semantics. This architectural choice will be further analyzed in the ablation studies.

### 3.2. Injection Mechanism of SAPE

We introduce several refinements to SAPE's injection mechanism to ensure its optimal function in the backbone.

**Shallow Injection Strategy** SAPE explicitly modulates token-wise information exchange, strengthening local feature alignment between reference and target images. However, the empirical observations suggest that this mechanism is not equally suitable for all attention layers. When SAPE is applied to deep layers, the generated images often exhibit fragmented textures. This phenomenon arises from deeper layers being more responsible for ensuring global visual coherence and continuity. Therefore, we adopt a shallow injection strategy, where SAPE is only utilized within the first 19 double-stream attention layers of the FLUX backbone.

**Sub-patch Position Embedding** In models like FLUX, images are encoded into tokens at the patch level, with each corresponding to multiple pixels. A naive approach would involve using only the surface coordinate $\mathbf{s}$ of the center pixel within a patch to derive SAPE for each respective token, which limits its geometric resolution as a continuous pose descriptor. Following the approach of PE-Field (Bai et al., 2025), we implement a sub-patch position embedding scheme. As shown in Figure 2, 21 surface coordinates are uniformly sampled from three resolution levels within each patch. These coordinates are encoded as SAPEs and then assigned to different attention heads. Given that FLUX utilizes 24-head attention, 21 heads are modulated by these embeddings, while remaining 3 heads remain unmodulated. Experimental results demonstrate that this strategy significantly enhances the precision of pose control.

**Balanced Segment Partitioning** SAPE partitions the total dimension $d$ into segments of lengths $d_x, d_y$, and $d_z$ to represent the three spatial coordinates. In fact, the original 2D RoPE in FLUX, which identifies token planar positions within the image, employs a similar strategy. In its implementation, the 128-dimensional vector is divided into three non-uniform segments: the first 16 dimensions remain unused, while the subsequent two 56-bit segments encode the horizontal and vertical positions of the tokens, respectively. For SAPE, all three dimensions of the surface coordinates $\mathbf{s}$ are equally vital for pose description. We therefore introduce a balanced segment partitioning scheme, allocating $d_x = 42, d_y = 42$, and $d_z = 44$. The experiments show that despite the difference in partitioning, SAPE maintains a strong synergy with the 2D RoPE of FLUX, leading to further performance gains.

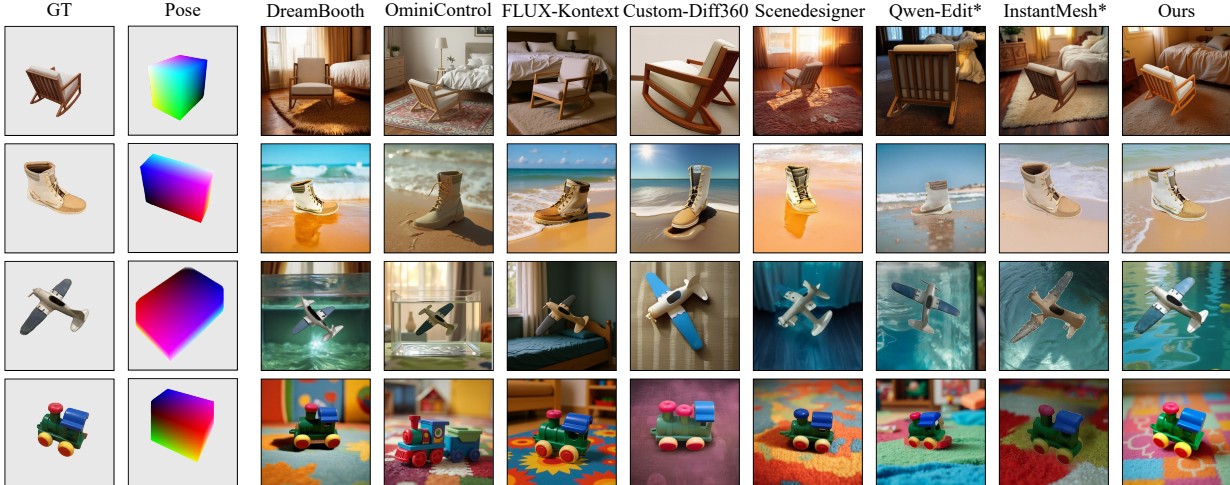

| GT | Pose | DreamBooth | OminiControl | FLUX-Kontext | Custom-Diff360 | Scenedesigner | Qwen-Edit* | InstantMesh* | Ours |

*Figure 3.* Qualitative comparison with state-of-the-art methods. We demonstrate generation results across diverse subjects and target poses. The top two rows feature samples from the 3D Assets set, while the bottom two rows present subjects from the Real-world set.

### 3.3. 3D-aware In-Context Learning

Our method is implemented on the multi-pose dataset. During training, we randomly sample a group of images and their corresponding poses from a subject's image set as references, denoted as $\{(I_{ref,i}, P_{ref,i})\}_{i=1}^{N}$. A separate pair $(I_{tgt}, P_{tgt})$ is selected to serve as the target. For simplicity, the text prompts are omitted in our formulation.

All images are encoded into latent tokens $\mathbf{z_{tgt}}$ and $\mathbf{z_{ref,i}}$ via VAE and concatenated into a single sequence. While the target image is positioned at the beginning of the sequence, the reference images follow in no specific order. The pose is converted into $E_{SAPE}$ and applied to all images, including the target. Notably, SAPE operates exclusively within subject regions, where the original 2D RoPE of FLUX remains concurrently in effect. The model is supervised by the rectified flow loss, consistent with the FLUX backbone:

$$\mathcal{L} = \mathbb{E}_{t,\mathbf{z_0},\mathbf{z_1},\epsilon} \left[ \|\mathbf{v}_\theta(\mathbf{z}_t, t, \mathbf{c}) - (\mathbf{z_1} - \mathbf{z_0})\|^2 \right] \quad (5)$$

where $\mathbf{z_1}$ represents the ground-truth latent, $\mathbf{z_0}$ is the gaussian noise, $\mathbf{z}_t = t\mathbf{z_1} + (1-t)\mathbf{z_0}$ is the noisy latent at timestep $t \in [0,1]$. $\mathbf{v}_\theta$ is the predicted velocity conditioned on the reference set and the target pose, collectively denoted by $\mathbf{c}$.

Our framework is entirely tuning-free, requiring no subject-specific optimization at test time. During sampling, users simply provide a reference set of the customized subject and a target pose. The model then performs multi-step denoising starting from pure gaussian noise following the diffusion schedule. SAPE is applied across all timesteps of the diffusion process to maintain precise pose control throughout the generation. A significant advantage of our approach is its ability in reference scaling. Although the model is trained on groups of 1 to 4 reference images, it

can seamlessly accommodate a larger number of references during inference. We observe that as the number of reference images increases, the appearance consistency of the generated subject improves continuously, demonstrating the robustness of our 3D-aware in-context learning paradigm.

## 4. Experiments

In this section, we first introduce our custom dataset and the experimental settings. Next, we present qualitative and quantitative comparisons against state-of-the-art methods. Finally, we provide in-depth analyses, including ablation studies, the 3D-Aware local feature interaction driven by SAPE, and the abilities of our method in reference scaling and continuous pose control.

### 4.1. Dataset

To facilitate research in this field, we curated a task-specific dataset featuring dense pose coverage and diverse scenes. This was necessitated by the limitations of existing resources: standard multi-pose datasets like CO3D (Reizenstein et al., 2021) are often restricted to monotonous scenes, potentially biasing background synthesis, while diverse synthetic datasets (Tan et al., 2025) suffer from extreme pose scarcity that hinders effective multi-reference training.

**Data Construction** The proposed dataset encompasses both 3D assets and real-world subjects. The 3D assets is sourced from the ABO (Collins et al., 2022) and GSO (Downs et al., 2022). We rendered high-quality 3D models across 36 randomly distributed poses. The real-world data is derived from the CO3D dataset (Reizenstein et al., 2021), focusing on rigid object categories that provide the necessary multi-pose images and camera parameters. Crucially, for all

*Table 1.* Quantitative comparison with state-of-the-art methods. We benchmark performance alongside support for multiple references and tuning-free inference. **Bold** and underlined indicate best and second-best results. * indicates pipeline-based methods.

| METHOD | MULTI-REFERENCES | TUNING-FREE | 3D-ASSETS TEST SET | | | | | REAL-WORLD TEST SET | | | | |
|---|---|---|---|---|---|---|---|---|---|---|---|---|
| | | | POSE ↓ | C-I ↑ | D-I ↑ | C-T ↑ | FID ↓ | POSE ↓ | C-I ↑ | D-I ↑ | C-T ↑ | FID ↓ |
| OMINICONTROL | ✗ | ✓ | - | 0.811 | 0.568 | **0.295** | 72.31 | - | 0.794 | 0.502 | 0.321 | 82.04 |
| FLUX-KONTEXT | ✓ | ✓ | - | 0.833 | 0.642 | 0.292 | 75.79 | - | 0.823 | 0.637 | 0.329 | 82.77 |
| DREAMBOOTH | ✓ | ✗ | - | 0.845 | 0.658 | 0.291 | 53.31 | - | 0.847 | 0.680 | 0.330 | 49.00 |
| CUSTOM-DIFF360 | ✓ | ✗ | 33.23 | 0.785 | 0.513 | 0.292 | 69.97 | 32.55 | 0.763 | 0.504 | 0.302 | 67.14 |
| SCENEDESIGNER | ✓ | ✗ | 25.70 | 0.826 | 0.605 | 0.292 | 63.06 | 23.88 | 0.837 | 0.641 | 0.328 | 54.21 |
| QWEN-EDIT* | ✗ | ✓ | 31.84 | 0.832 | 0.592 | 0.290 | 49.18 | 39.49 | 0.826 | 0.591 | 0.331 | 61.14 |
| INSTANTMESH* | ✓ | ✓ | **7.62** | 0.870 | **0.766** | 0.285 | 49.83 | 16.45 | 0.817 | 0.609 | 0.320 | 70.24 |
| **OURS** | ✓ | ✓ | 10.75 | **0.875** | 0.758 | 0.293 | **48.65** | **13.20** | **0.867** | **0.758** | **0.332** | **47.93** |

subjects, we computed the oriented bounding box to serve as the volumetric bounding box, which forms the basis of our SAPE mechanism. Finally, to enhance generalizability, a specialized generative pipeline is employed to synthesize diverse and complex backgrounds for our data. Further implementation details are provided in the Appendix A.

**Statistics and Data Split** The final dataset comprises 7,665 unique objects across 371 categories, totaling 153,227 images. Specifically, the **3D Assets subset** contributes 6,464 objects from 367 categories, amounting to 68,549 images. The **Real-world subset** consists of 1,201 objects distributed across 4 categories, providing 84,678 images. For evaluation, we curated a representative test set consisting of 35 objects from the 3D assets subset and 20 objects from the real-world subset. Each test object is associated with at least 36 distinct poses. During comparative experiments, all methods are required to generate images corresponding to these specific poses. In total, 2,700 images were generated for the final evaluation, comprising 1,260 images from the 3D assets data and 1,440 images from the real-world data.

### 4.2. Experimental Settings

**Implementation Details** Our framework is built upon the FLUX.1-dev text-to-image diffusion model. All input images are resized to a resolution of $512 \times 512$. During training and inference, we randomly sample up to $N = 4$ reference images per iteration. For parameter-efficient fine-tuning, we employ Low-Rank Adaptation (LoRA) (Hu et al., 2022) injected into all linear projections of the transformer. The LoRA rank and alpha are both set to 16, with weights initialized using a gaussian distribution. The model is optimized using the prodigy optimizer (Mishchenko & Defazio, 2024) with an adaptive learning rate strategy, initialized at a learning rate of 1.0 and a weight decay of 0.01. The training spans a total of 100,000 steps with a batch size of 4. All experiments are conducted on four NVIDIA L40s.

**Metrics** We quantitatively evaluate our method across three

aspects. For **Pose Accuracy**, we adopt Orient Anything (Wang et al., 2025) to estimate the pose of generated subjects. As concluded in their original study, this metric is proven to be robust and effective for evaluating generated images. The Mean Absolute Error (MAE) averaged across azimuth, elevation, and rotation between the generated and target poses is reported. For **Identity Consistency**, we compute the pairwise feature similarity between the generated and ground truth images, utilizing both CLIP-I (ViT-B/32) and DINO-I (ViT-S/16). Regarding **Text Alignment**, we employ CLIP-T, which measures the cosine similarity between the features of the generated image and the text prompt. Finally, FID is used to evaluate overall **Image Quality**.

### 4.3. Comparison with Existing Methods

We benchmark Pose-ICL against seven state-of-the-art methods across three categories. First, for **Subject Customization**, we compare against DreamBooth (Ruiz et al., 2023), OminiControl (Tan et al., 2025), and FLUX-Kontext (Black Forest Labs et al., 2025). Since these methods lack explicit 3D control mechanisms, they are excluded from the quantitative pose accuracy evaluation. Second, for **Pose-Controllable Generation**, we evaluate Custom Diffusion 360 (Kumari et al., 2024) and SceneDesigner (Qin et al., 2025). Finally, for **Editing-based Pipelines**, we include the pipelines that mainly based on Qwen-Edit (Wu et al., 2025) and InstantMesh (Xu et al., 2024). Qualitative and quantitative results are presented in Figure 3 and Table 1, with implementation details of these methods provided in the Appendix B.

**Comparison with Subject Customization Methods** The results in Table 1 demonstrate our method's superiority across two datasets. In terms of identity preservation, we significantly outperform the strongest competitor, Dream-Booth, by margins of 0.1 and 0.078 in DINO-I scores. This performance gain highlights the effectiveness of our 3D-aware learning paradigm. By interpreting subjects through a structured 3D perspective, the model can aggregate spa-

*Table 2.* Ablation study for the key components. "Shallow Inj.", "Sub-patch PE", and "Balanced Seg." denote Shallow Injection Strategy, Sub-patch Position Embedding, and Balanced Segment Partitioning. **Bold** and underlined indicate best and second-best results.

| SETTINGS | SHALLOW INJ. | SUB-PATCH PE | BALANCED SEG. | 3D-ASSETS TEST SET | | | | | REAL-WORLD TEST SET | | | | |
|---|---|---|---|---|---|---|---|---|---|---|---|---|---|
| | | | | POSE ↓ | C-I ↑ | D-I ↑ | C-T ↑ | FID ↓ | POSE ↓ | C-I ↑ | D-I ↑ | C-T ↑ | FID ↓ |
| BASELINE | - | - | - | 22.88 | 0.861 | 0.721 | 0.292 | 51.87 | 29.97 | 0.856 | 0.720 | 0.329 | 50.17 |
| + SAPE | ✗ | ✗ | ✗ | 19.33 | 0.830 | 0.661 | 0.290 | 60.75 | 23.72 | 0.835 | 0.661 | 0.325 | 67.03 |
| + SHALLOW INJ. | ✓ | ✗ | ✗ | 14.12 | 0.859 | 0.719 | 0.292 | 54.29 | 16.71 | 0.859 | 0.727 | 0.330 | 69.20 |
| + SUB-PATCH PE | ✓ | ✓ | ✗ | 16.11 | 0.862 | 0.722 | **0.296** | 54.78 | 15.78 | 0.862 | 0.734 | 0.331 | 56.33 |
| + BALANCED SEG. | ✓ | ✓ | ✓ | **10.75** | **0.875** | **0.758** | 0.293 | **48.65** | **13.20** | **0.867** | **0.758** | **0.332** | **47.93** |

tial information from multiple viewpoints more efficiently, ensuring that the subject's appearance remains robustly consistent across diverse target poses.

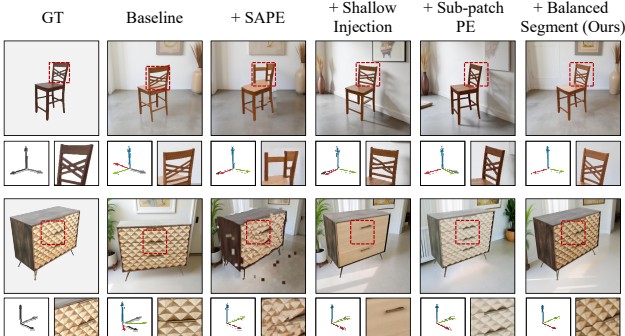

*Figure 4.* Qualitative ablation study. We visualize object pose axes and highlight key features for clearer comparison.

**Comparison with Pose-Controllable Generation Methods** The quantitative results in Table 1 confirm the superior precision of our framework in pose-controllable generation. On 3D assets, our method achieves a pose error of 10.75, which markedly outperforms the 33.23 recorded by Custom-Diff360 and the 25.70 reported for SceneDesigner. These results underscore the advantages of our unified, tuning-free design. On one hand, it avoids the complexities inherent in joint 3D-2D optimization. On the other, by treating pose control and customization as mutually reinforcing rather than independent tasks, Pose-ICL achieves deeper feature integration, ensuring precise alignment and high-fidelity identity preservation.

**Comparison with Editing-based Pipelines** Multi-stage approaches exhibit distinct limitations. Qwen-Edit, relying primarily on text prompts, lacks the necessary granularity for precise and continuous perspective control. Meanwhile, the reconstruction quality of InstantMesh is heavily contingent on the accuracy of reference poses. In real-world scenarios, inevitable estimation errors severely degrade both image fidelity and subject identity, as exemplified by the distorted toy plane in Figure 3. These observations are quantitatively confirmed in Table 1, where InstantMesh lags behind our method by a substantial margin of 0.149 in the real-world

DINO-I score. This gap highlights our framework's superior robustness against pose estimation noise.

### 4.4. Ablation Study

We conduct a progressive ablation study across five settings: **(1) Baseline** concatenating surface coordinate maps $S$ with reference images to serve as the visual input context, rather than employing SAPE; **(2) + SAPE** feeding surface coordinates as SAPE; **(3) + Shallow Inj.** introducing Shallow Injection Strategy; **(4) + Sub-patch PE** introducing Sub-patch Position Embedding; and **(5) + Balanced Seg. (Ours)** introducing Balanced Segment Partitioning. Quantitative and qualitative results are provided in Table 2 and Figure 4.

**Ablation for SAPE** Quantitative results indicate that incorporating SAPE significantly reduces Pose by 3.55 and 6.25 across the two datasets. By utilizing SAPE as the primary input vehicle, we allow 3D coordinates to influence features indirectly through attention modulation. This architectural choice effectively prevents non-image signals from directly interfering with the subject's visual semantics, and bridging the gap between abstract volumetric information and the latent embedding space. As shown in Figure 4, this leads to substantially improved pose accuracy, particularly for geometrically complex subjects like chairs and cabinets.

**Ablation for Shallow Injection Strategy** By restricting SAPE to the shallow attention layers, the framework leverages these layers to retrieve aligned patch features, while the deeper layers are preserved as visual refiners. This design ensures high-fidelity texture synthesis while maintaining precise control, as evidenced by the superior DINO-I, CLIP-I, and FID scores compared to the global injection variant.

**Ablation for Sub-patch Position Embedding** This module allows specific attention heads to process spatial interactions at higher resolutions, enabling the model to faithfully capture intricate structures, such as the chair back in Figure 4, that are otherwise difficult to represent. This enhancement leads to consistent improvements in DINO-I and CLIP-I, demonstrating its effectiveness in maintaining subject-specific attributes.

**Ablation for Balanced Segment Partitioning** By redis-

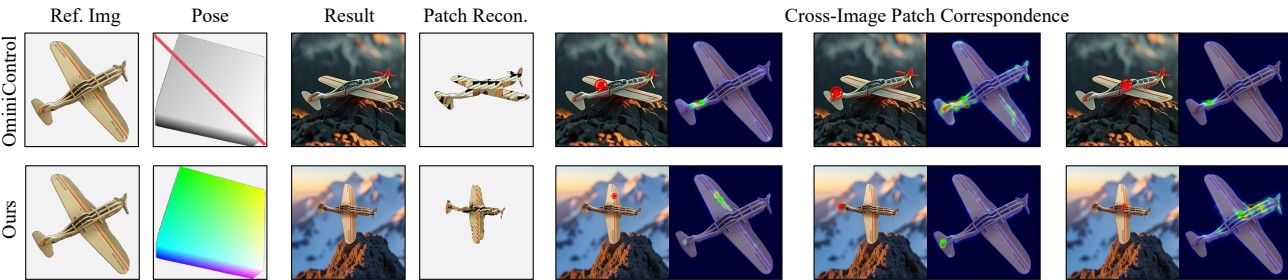

*Figure 5.* 3D-Aware Local Feature Interaction. In cross-Image patch correspondence map, the red dot indicates a specific patch in the generated image, while the green dot marks the corresponding patch in the reference image with the highest attention activation.

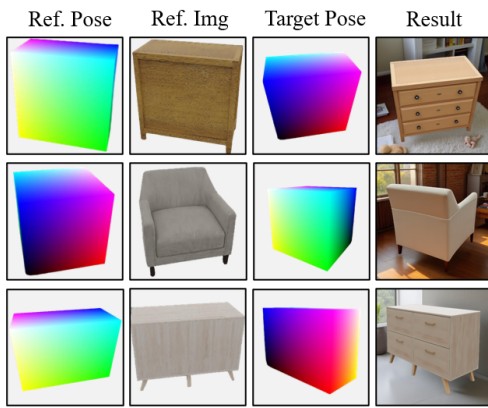

*Figure 6.* Generation results under extreme pose discrepancy.

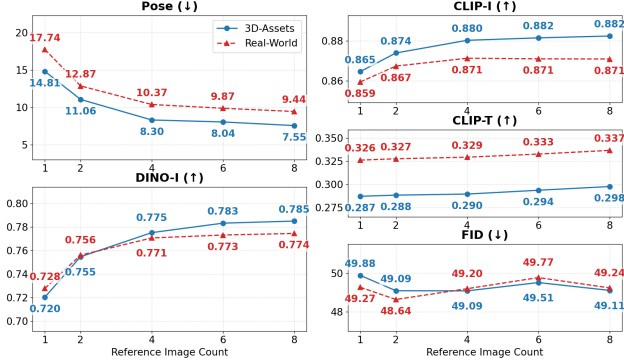

*Figure 7.* The quantitative evaluation of reference scaling.

tributing FLUX's default dimension allocations to treat azimuth, elevation, and roll equally, this component facilitates stricter alignment with ground-truth poses. Quantitatively, this refinement reduces Pose MAE by 5.36 on the Realworld dataset and 2.58 on the Synthetic dataset, achieving the optimal performance reported in our final framework.

### 4.5. 3D-Aware Local Feature Interaction

To examine the SAPE's effectiveness in promoting 3D-aware local feature interaction, we visualize the correspondence between the generated and reference patches in the Double-Stream attention layer through Attention-Guided Patch Reconstruction. As shown in the fourth column of Figure 5, our framework maintains strict 3D-aware local interactions, enabling the precise retrieval of fine-grained details such as wing stripes and tail structures. By leveraging SAPE to anchor interactions within a coherent volumetric space, the model ensures that initial layers aggregate spatially consistent features. This structural precision allows subsequent single-stream blocks to function as effective visual refiners, yielding superior feature fidelity and accurate pose alignment. While paradigms lacking explicit spatial guidance often suffer from disorganized patch retrieval, as observed in the OminiControl baseline, our approach facilitates a direct and robust feature exchange that preserves

subject identity throughout the generation process.

Beyond local patch correspondence, this explicit 3D awareness also enables the model to handle extreme view changes. For instance, as shown in Figure 6, even when the reference image only captures the back of a cabinet, the model successfully infers the spatial requirements and renders a plausible front view with drawers, demonstrating true volumetric understanding over 2D pattern matching.

### 4.6. Reference Scaling and Continuous Pose Control

**Reference Scaling** Leveraging the scalable DiT architecture, we evaluate the model with $N \in \{1, 2, 4, 6, 8\}$ reference images. As shown in Figure 7, increasing $N$ yields consistent improvements in pose accuracy and identity fidelity, while FID remains relatively stable. This performance trajectory suggests that Pose-ICL effectively aggregates multi-pose information to refine its internal 3D representation.

**Continuous Pose Control** We further assess the smoothness of pose transitions by fixing reference images and systematically interpolating camera angles. Figure 8 demonstrates coherent geometric transitions across varying Azimuth ($\phi$) and Elevation ($\theta$) at $10°$ intervals. The consistent appearance throughout these sequences underscores the model's robust volumetric understanding, surpassing the mere memorization of discrete views.

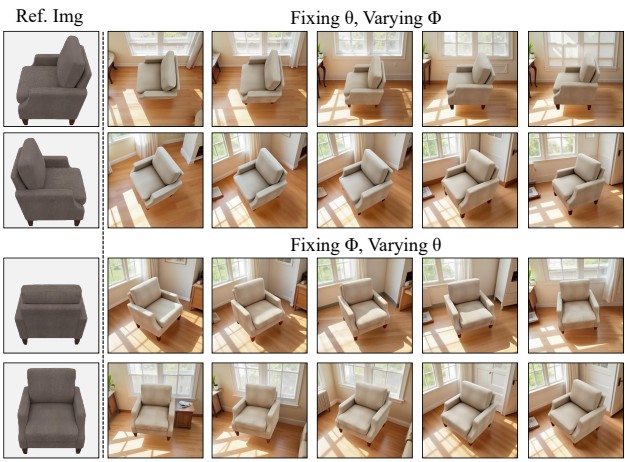

*Figure 8.* Generation results under continuous pose control.

## 5. Conclusion

This paper introduces Pose-ICL, a tuning-free, 3D-aware in-context learning paradigm for pose-controllable subject customization. By leveraging SAPE, our framework achieves a volumetric understanding that bridges the gap between 3D geometry and 2D generation. Evaluations demonstrate superior performance in pose accuracy and identity fidelity over existing methods. By anchoring generation in 3D space, we move beyond mere pattern imitation toward a more structured and geometrically-grounded form of subject customization.

## Acknowledgements

This work was supported in part by the the National Key R&D Program of China under Grant No. 2024YFB3311801, the National Natural Science Foundation of China under Grant No. 62473290, the Shanghai Municipal Science and Technology Commission under Grant No. 25511102800, the Shanghai Municipal Commission of Economy and Information Technology under Grant No. 2024-GZL-RGZN-01008.

## Impact Statement

This paper presents work whose goal is to advance the field of Machine Learning. There are many potential societal consequences of our work, none which we feel must be specifically highlighted here.

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

# A. Dataset Construction

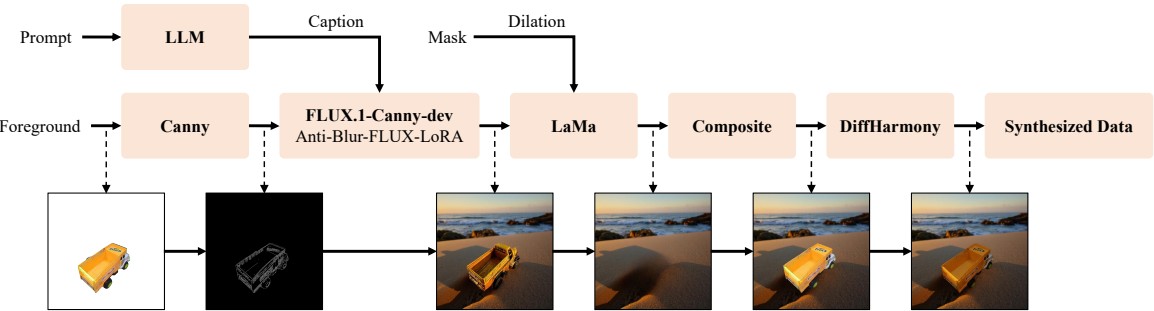

*Figure 9.* Automated Data Synthesis Pipeline. We leverage FLUX and LoRA to generate diverse backgrounds, utilizing LaMa for background cleaning and DiffHarmony for final image harmonization.

## A.1. Data Sourcing and Volumetric Processing

**Data Sources** Our training dataset comprises two distinct categories: 3D synthetic assets and real-world subjects.

- *3D Assets:* We source high-quality 3D models from the ABO (Collins et al., 2022) and GSO (Downs et al., 2022) datasets. These provide precise geometry and texture information, serving as the ground truth for our work.

- *Real-world Subjects:* We utilize the CO3D (Reizenstein et al., 2021) dataset, which provides multi-pose image sequences of real-world subjects. In addition to raw images, the dataset offers sparse point clouds and camera parameters (extrinsics and intrinsics) for each frame. In our setup, we exploit the equivalence of relative motion, treating camera poses as object poses relative to the scene center. Four categories are used in this work, including toy plane, toy bus, toy train and toy truck.

**Data Processing and Bounding Box Rendering** We employ specific processing pipelines for each category:

- *multi-pose Image Acquisition:* For 3D assets, we sample multi-pose images by rendering the models in a render software, obtaining clean foregrounds with transparent backgrounds. For real-world subjects, we implement a strict filtering protocol: images are first screened for blurriness, and samples with incomplete object visibility (e.g., truncated by image borders) are discarded to ensure valid supervision.

- *Volumetric Bounding Box Rendering:* A core component of our method is the projected Volumetric Bounding Box image. For 3D assets, this is trivial; we render the bounding box directly alongside the foregrounds using the rendering engine. For real-world subjects, where explicit boxes are absent, we derive them from the provided point clouds. Specifically, we compute the minimal oriented bounding box of the object's point cloud to align the coordinate system with its principal axes. We then construct the mesh representations of the box faces and render them onto the image plane using the corresponding camera intrinsics and extrinsics. By employing barycentric interpolation during rasterization, we generate a dense, colored condition map that encodes the box's spatial structure.

## A.2. Automated Data Synthesis Pipeline

Since the raw 3D renders lack backgrounds and the CO3D images often contain monotonous environments, we constructed an pipeline to enhance the visual quality and diversity of images. As illustrated in Figure 9, the pipeline operates as follows:

1. *Prompt Generation:* An LLM is employed to generate descriptive scene captions based on the object category.

2. *Background Hallucination:* We utilize FLUX.1-Canny-dev equipped with an Anti-Blur LoRA. It takes the foreground's Canny edge map and the generated caption as input to synthesize a coherent scene.

3. *Clean Background Extraction:* To avoid artifacts where the generated object does not perfectly align with the original foreground, we dilate the foreground mask and use LaMa (Suvorov et al., 2022) to inpaint the foreground area, producing a clean background plate.

4. *Harmonization:* Finally, the original high-quality foreground is composited onto the generated background. We apply DiffHarmony (Zhou et al., 2024) to adjust the lighting and color tones, ensuring a photorealistic blend between the subject and the new environment.

## B. Implementation Details of Comparison Methods

We reproduced seven baseline methods for comparison. In this section, we provide the codebases, checkpoints, and specific implementation details for each.

DreamBooth (Ruiz et al., 2023) (*https://github.com/huggingface/diffusers*): We used FLUX.1-dev as the backbone. For each subject, we trained a LoRA with a rank of 4 for 1800 optimization steps using 8 images. Sampling was performed using 30 inference steps and a guidance scale of 3.5.

OminiControl (Tan et al., 2025) (*https://github.com/Yuanshi9815/OminiControl*): This method used FLUX.1-schnell as the backbone. We utilized the *subject_512* checkpoint. This approach is restricted to a single reference image, and we followed the recommended sampling parameters: the guidance scale was set to 3.5, and the number of inference steps was set to 8.

FLUX-Kontext (Black Forest Labs et al., 2025) (*https://github.com/huggingface/diffusers*): We utilized the FLUX.1-Kontext-dev checkpoint within the Diffusers library. Conditioned on the provided reference inputs, we followed the recommended sampling parameters: the guidance scale was set to 2.5, and the number of inference steps was set to 28.

Custom-Diff360 (Kumari et al., 2024) (*https://github.com/customdiffusion360/custom-diffusion360*): This method used SDXL as the backbone, and required subject-specific tuning. We performed optimization for 1600 steps per subject using 8 images. For sampling, we used the Euler scheduler for 50 steps, with an image guidance scale of 3.5 and a text guidance scale of 7.5.

SceneDesigner (Qin et al., 2025) (*https://github.com/FudanCVL/SceneDesigner*): This method used SD3.5 as backbone and incorporates DreamBooth for subject learning. We performed optimization for 1800 steps per subject using 8 images. For sampling, we used the FlowMatchEuler scheduler for 20 steps, with conditions injected only during the first 15 steps and a text guidance scale of 7.5.

Qwen-Edit Pipeline (Wu et al., 2025) (*https://github.com/huggingface/diffusers*): This pipeline utilizes a checkpoint specifically fine-tuned for camera pose control (*https://huggingface.co/dx8152/Qwen-Edit-2509-Multiple-angles*). Starting from a single reference image, the pipeline undergoes three sequential instruction-based edits: (1) **Azimuth adjustment** e.g., "Rotate the camera degrees to the right/left", (2) **Elevation adjustment** e.g., "Move the camera downwards/upwards for degrees", and (3) **Background synthesis** e.g., "Fill the background with...". The instructions were drafted in accordance with the recommended templates. Since camera movement is equivalent to subject rotation in a static scene, we performed the necessary coordinate transformations. Sampling was conducted using FlowMatchEuler for 50 steps, with a True CFG scale of 4.0 and a guidance scale of 1.0.

InstantMesh Pipeline (Xu et al., 2024) (*https://github.com/TencentARC/InstantMesh*): This two-stage pipeline consists of a tuning-free reconstruction and background synthesis. (1) **Reconstruction** we used the *instant_nerf_large* checkpoint to generate a NeRF-based representation using 4 real reference images (its maximum capacity). The object was rendered at resolution of 512. (2) **Background synthesizing** This stage wes handled by the Qwen-Edit pipeline using the parameters mentioned above.

## C. Additional Results

In this section, we present further analyses to demonstrate the robustness and versatility of our method.

### C.1. Robustness to Scale Variations

In DiT architectures, images are processed as sequences of patch embeddings rather than pixel-wise grids. Due to this inherent spatial discretization, a critical question arises: can the model effectively interpret pose signals when the projected volumetric bounding box occupies only a small fraction of the canvas (i.e., represented by very few tokens)? To investigate this, we evaluate our method as the camera distance increases. As illustrated in Figure 10, even as the subject's spatial footprint significantly diminishes, our method maintains precise pose alignment and consistent appearance details. This demonstrates that SAPE effectively injects geometric guidance even at low spatial resolutions.

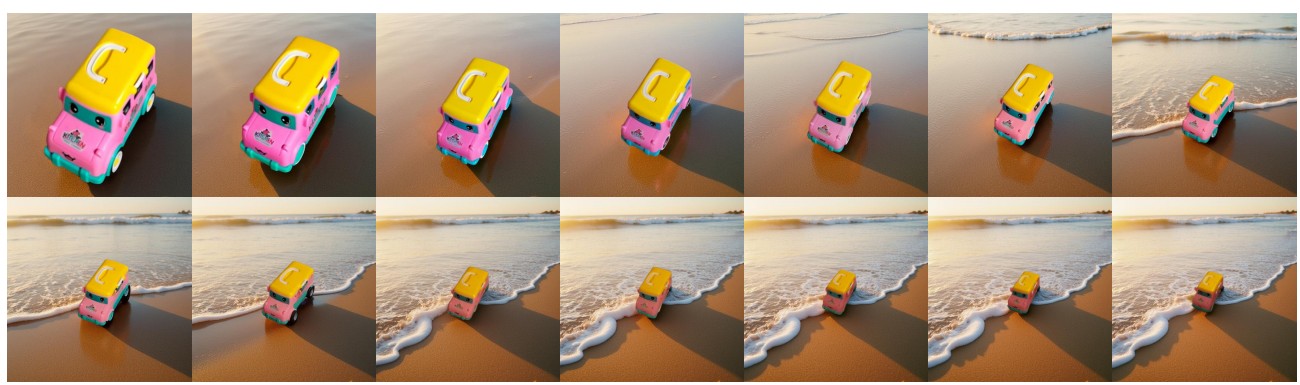

Distance

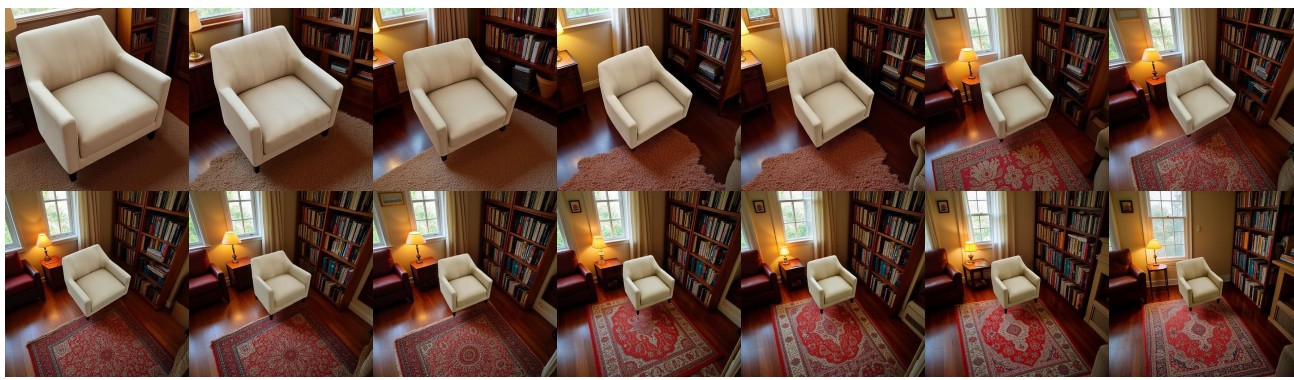

Distance

*Figure 10.* Evaluation of robustness to scale variations. We gradually increase the camera distance to reduce the subject's spatial footprint (occupancy) in the image. Despite the diminishing resolution of the volumetric condition, our method maintains high fidelity and precise pose alignment, demonstrating the effectiveness of SAPE even for small objects.

### C.2. Quantitative Analysis of Reference Pose Count

In this experiment, we utilize a single 3D asset with fixed remaining viewing parameters and sample 36 target poses at $10°$ intervals across the full $360°$ azimuth ($\theta$). We perform image generation conditioned on five distinct, deterministically sampled reference sets ($N \in \{1, 2, 4, 6, 8\}$) and visualize the pose error curves as a function of $\theta$. To rigorously assess the impact of geometric coverage, the reference angles are fixed as follows:

- $N = 1$: Fixed at $\theta = 0°$.
- $N = 2$: Fixed at $\theta = \{0°, 180°\}$.
- $N = 4$: Fixed at $\theta = \{0°, 90°, 180°, 270°\}$.
- $N = 6$: Fixed at $\theta = \{0°, 60°, 120°, 180°, 240°, 300°\}$.
- $N = 8$: Fixed at $\theta = \{0°, 50°, 90°, 140°, 180°, 230°, 270°, 320°\}$.

As illustrated in Figure 11(a), the single-reference setting ($N = 1$) exhibits the highest instability, characterized by two prominent error peaks and an overall average error of $13.63°$. As the number of reference images increases, the error curve progressively flattens, and the mean error steadily decreases. Notably, the performance achieves stability at $N = 4$. While increasing the reference count to $N = 6$ and $N = 8$ yields further minor reductions, the marginal gains diminish significantly. Consequently, to balance accuracy with computational efficiency, we determine that $N = 4$ represents the optimal configuration.

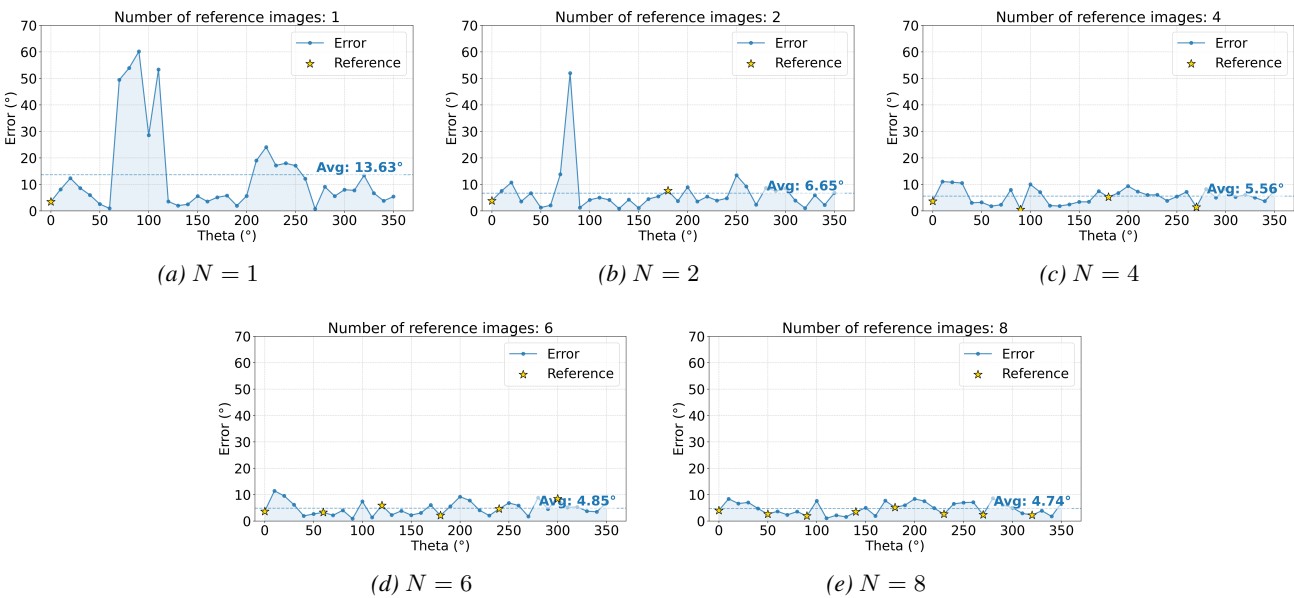

*Figure 11.* Pose estimation error vs. Azimuthal Angle $\theta$. Yellow stars indicate the positions of fixed reference images. The error curves show that increasing geometric coverage from $N = 1$ to $N = 4$ effectively eliminates error spikes caused by blind spots. However, further increasing the density to $N = 6$ or 8 yields diminishing returns, confirming $N = 4$ as the optimal configuration.

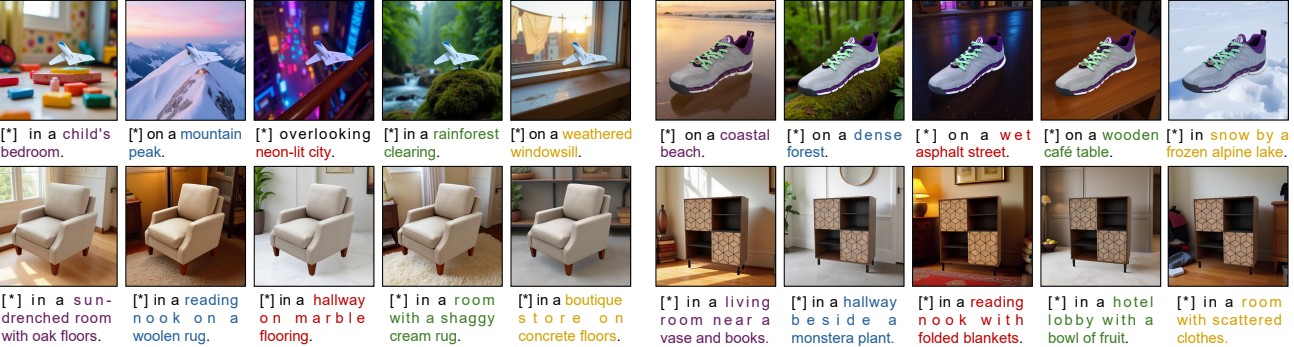

*Figure 12.* The generation results of the our method under different text prompts.

## C.3. Background Controllability

We evaluated the model's ability to decouple subject customization from scene generation using complex prompts. As visualized in Figure 12, Pose-ICL successfully synthesizes realistic backgrounds semantically aligned with the text. Crucially, despite dramatic environmental variations, the model maintains high fidelity and precise volumetric control for the foreground objects. This confirms that our method effectively contextualizes subjects without compromising geometric accuracy.

## C.4. Inference Pipeline in Practice

In practical applications ("in the wild"), ground truth 3D bounding boxes are not available. To address this, we designed an inference pipeline that adapts to the density of the input poses, as shown in Figure 13.

- *Pose and Point Cloud Estimation:*
  - *Dense Image Sequences:* When a video or a dense set of images is available, we utilize COLMAP (Schönberger & Frahm, 2016; Schönberger et al., 2016) method, to accurately reconstruct the sparse point cloud and estimate camera parameters.

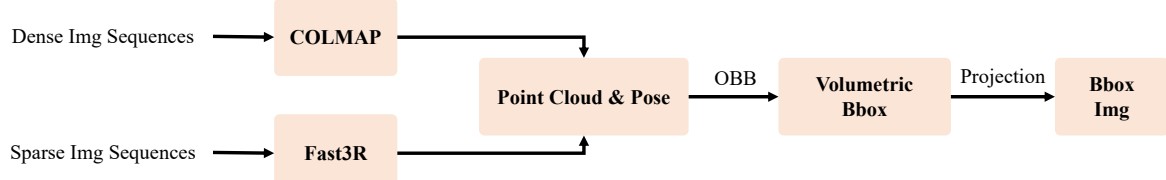

*Figure 13.* Inference Pipeline in practice. Depending on the input data density, we employ either COLMAP (dense) or Fast3R (sparse) to recover the underlying 3D structure, which is then processed to get the required Volumetric Bounding Box conditions.

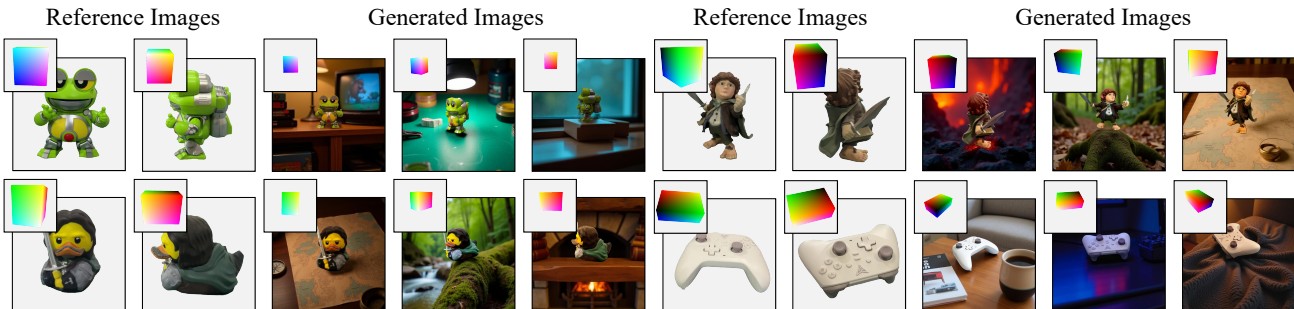

*Figure 14.* The generation results of the our method in practice.

> – *Sparse Image Sequences:* For scenarios with only a few snapshots, we employ Fast3R (Yang et al., 2025), a 3D reconstruction model, to rapidly infer the point cloud and camera poses.

- *Condition Generation:* Once the point cloud and poses are obtained, we apply the same method used in our dataset construction. The point cloud is aligned to define the Volumetric Bounding Box, which is then projected to the target pose to serve as the structural condition for Pose-ICL.

Qualitative results in Figure 14 confirm that Pose-ICL generalizes effectively in practice, achieving high-quality pose-controllable personalization beyond synthetic datasets.

## C.5. Robustness to Noisy Pose Annotation

*Table 3.* Quantitative evaluation of robustness to noisy pose annotations. The model maintains stable identity fidelity (DINO-I) even when subjected to significant pose noise.

| Pose Noise | POSE $\downarrow$ | DINO-I $\uparrow$ |
|---|---|---|
| $0°$ | 2.18 | 0.777 |
| $\pm 10°$ | 6.73 | 0.767 |
| $\pm 20°$ | 11.05 | 0.752 |
| $\pm 30°$ | 16.67 | 0.742 |

To investigate the model's robustness against pose estimation errors, we conducted additional experiments by injecting varying degrees of noise ($0°$, $\pm 10°$, $\pm 20°$, $\pm 30°$) into the pose annotations of one randomly selected reference image. We report the pose error (**POSE**) and DINO similarity (**DINO-I**) across 10 test objects (sampling 15 images per noise level) to evaluate control accuracy and identity fidelity, respectively.

As can be seen, while increased pose noise naturally leads to higher pose error (**POSE**), the identity fidelity (**DINO-I**) remains relatively stable. This robustness is corroborated by both quantitative metrics and qualitative observations, as shown in Table 3 and Figure 15.

We attribute this robustness to the feature interaction mechanism of the attention layers. As formulated in Equation (4), the attention score between two features is jointly modulated by the SAPE and the dot product of query $q$ and key $k$. When

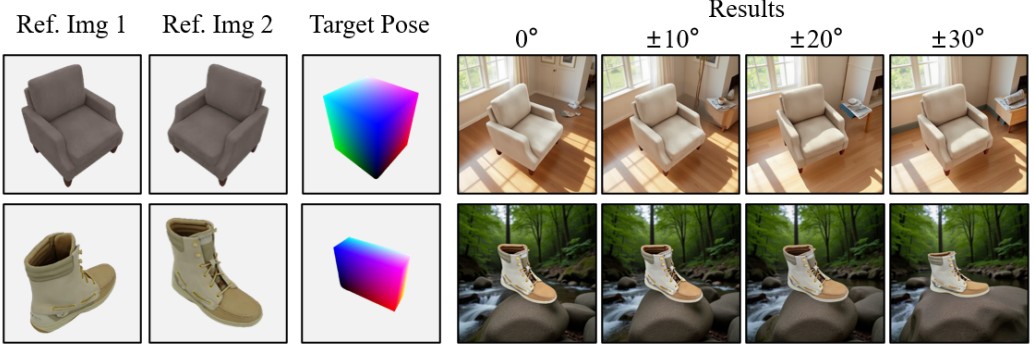

*Figure 15.* Generation results under different levels of pose annotation noise.

two reference features $a'$ and $b'$ provide conflicting information due to noisy pose annotations, the generated feature $c'$ will preferentially attend to the feature with the higher score rather than mechanically blending them. This competitive mechanism ensures that the model maintains structural integrity even under imperfect pose guidance.

## D. Failure Case and Future Work

Despite the robust pose control achieved by our method, we occasionally observe a lack of visual harmony between the foreground subject and the background, which compromises the photorealism of the generated images. This issue stems primarily from our data construction pipeline. Since the training samples were created by compositing objects onto synthetically generated backgrounds, this process inevitably compromised the harmonious interaction between the foreground and the background. Consequently, the model inadvertently learned this composition bias during training, leading to degradation in the coherence of inference results.To address this, a promising direction is to transition towards a fully generative data paradigm. Instead of relying on composition pipelines, we aim to leverage the inherent capabilities of large-scale foundation models to directly synthesize complete, holistic scenes where the subject and background are generated simultaneously. By ensuring that the foreground and environment are generated in a single pass, we can effectively enhance the overall integrity the training data.

