# OpenReview forum: "Pose-ICL: 3D-Aware In-Context Learning for Pose-Controllable Subject Customization"
_ICML.cc/2026/Conference — ICML 2026 regular_

### Official Review · Reviewer_mjGw · 2026-03-11

**Soundness:** 3
**Presentation:** 3
**Significance:** 3
**Originality:** 3
**Overall Recommendation:** 4
**Confidence:** 4

**Summary:**

This paper proposes Pose-ICL, which utilizes tools like COLMAP or Fast3R to estimate pose and 3D bounding boxes. By employing the 3D bounding box as a bridge and modifying the position encoding of DIT, it elevates the 2D image generation task to the level of 3D spatial perception. This approach seeks to strike a balance between “efficiency (no fine-tuning)” and “accuracy (3D consistency).”

**Compliance With Llm Reviewing Policy:**

Affirmed.

**Final Justification:**

My concerns have been resolved.  I maintain my original rating.

**Key Questions For Authors:**

1. How does image quality degrade when COLMAP or Fast3R provide inaccurate 3D poses or scales (especially under sparse reference views)?

2. Input requirements appear stricter now. Can existing quality be maintained if reference images are randomly captured under varying scenes and lighting conditions?

3. Can the 3D bounding box position contradict the semantic intent specified in the text prompt? For example, if the prompt requires the object to appear in a corner, will the 3D bounding box adaptively position itself in the corner?

4. Given that bounding boxes possess “axis-independence,” how does the SAPE mechanism ensure consistent front-facing orientation for asymmetrical objects?

**Limitations:**

yes

**Strengths And Weaknesses:**

Strengths

1.Unlike methods such as DreamBooth or Custom Diffusion that require fine-tuning for each new object, Pose-ICL employs an In-Context Learning (ICL) paradigm. It achieves direct generation by inputting reference images and poses during inference, significantly enhancing efficiency.

2.By embedding 3D geometric constraints directly into positional encodings via SAPE, the model comprehends objects as 3D volumes rather than simple 2D patterns. This resolves the common “cross-view appearance inconsistency” issue found in existing customization methods.

3.Both metrics and visualizations demonstrate significant advantages of the proposed method over previous approaches.

Weaknesses

1.Performance heavily relies on the accuracy of initial pose estimation. What happens if COLMAP or Fast3R fail to accurately reconstruct point clouds or estimate poses? Point cloud reconstruction errors are more likely with fewer reference images; demonstrating results with dimensional estimation errors would be beneficial.

2.Figure 3 reveals a lack of visual harmony between generated objects and backgrounds, appearing more rigid and less realistic than FLUX-Kontext (acknowledged by the authors).

3.The 3D bounding box simultaneously determines object position. How is this bounding box positioned? Is it randomly placed or based on specific rules? Could this conflict with the user's intended prompt?

4.The logic behind the box is somewhat unclear. Given a box, is the generated object fixed in orientation? Or is it possible to generate objects rotated 180 degrees (while maintaining consistent bounding box dimensions)? The introduction mentions that the box is “axis-agnostic,” which is somewhat confusing.

---

> ### Author Rebuttal · Authors · 2026-03-31
>
> **Q1: Robustness to Noisy Pose Annotation**
>
> Please refer to our response to **#R1Q1 (9uwc)**.
>
> **Q2: Clarification about Diverse Ref. Images**
>
> Thank you for your professional review. Our model supports reference images captured under diverse scenes and lighting. This issue was already considered during the data preparation stage. As detailed in **Appendix A.1**, the background of each image was created independently under different prompts, and their foreground were harmonized to present varied lighting and shading. In fact, all results presented in the manuscript are generated from randomly sampled reference images with diverse scenes and lighting.
>
> **Q3: Clarification about 3D box’s Position**
>
> This is an interesting point. Since our method provides 6-DoF pose control, the position of the 3D box is explicitly determined by user-provided pose parameters and does not move adaptively. Our tests show that the 3D box provides stronger spatial constraints than the text prompt. In cases of conflict, the model prioritizes the 3D box. This is likely because base models like FLUX still have limitations in supporting fine-grained position control through text alone. We hope this clarifies your concern.
>
> **Q4: Clarification about "axis-agnostic"**
>
> We noticed that you also expressed similar confusion in the weakness part. As stated in **Section 3.1**, the 3D boxes in our framework are directional. Each axis is normalized to a range of $[0, 1]$, allowing the model to distinguish between opposite faces (e.g., front vs. back) and preventing $180^\circ$ flips.
>
> Regarding the "axis-agnostic", it refers to the fact that axes do not have a fixed, universal physical meaning across different object categories. For example, the $x=1$ plane might represent the "front" of a cabinet in one object, but the "top" in another. The only requirement is that the box-to-object binding should be consistent across the views of the same subject. This feature ensures that SAPE can handle asymmetrical objects without ambiguity, as demonstrated in **Fig. 13 (Objects 2 and 3)**. We will clarify this in the revised manuscript.

---

> > ### Author Rebuttal · Reviewer_mjGw · 2026-04-01
> >
> > I appreciate the authors’ response, which has given me a clearer understanding of the method’s capabilities.

---

### Official Review · Reviewer_AtXa · 2026-03-13

**Soundness:** 3
**Presentation:** 3
**Significance:** 2
**Originality:** 3
**Overall Recommendation:** 4
**Confidence:** 3

**Summary:**

This manuscript presents a 3D-aware in-context learning framework for tuning-free, pose-controllable subject customization. The authors propose Surface-Anchored Position Embedding (SAPE) and condition on multiple reference images to maintain texture consistency under pose changes. Specifically, the method first extracts an oriented volumetric bounding box of the subject and renders it into surface coordinate maps. These coordinates are then encoded with RoPE and integrated into the attention computation as positional embeddings. To make SAPE more effective and stable, the authors apply it only in shallow layers and introduce sub-patch positional encoding to improve pose-control precision. The authors also construct a dataset containing both 3D assets and real-world subjects to support training. With these designs, Pose-ICL significantly outperforms existing personalization and editing-based baselines in both pose accuracy and texture consistency.

**Compliance With Llm Reviewing Policy:**

Affirmed.

**Key Questions For Authors:**

Overall, based on the comments above, I believe the manuscript would be more complete with additional ablation studies and stronger comparisons to other pose-conditioning designs. At this stage, I lean toward a weak accept.

**Limitations:**

Refer to the weakness.

**Strengths And Weaknesses:**

Strength
1. The proposed framework is tuning-free and achieves strong performance compared to existing methods in pose accuracy and appearance preservation.

2. The contributed dataset could be valuable to the community, especially if it is open-sourced.

Weakness
1. Since SAPE is the core technical contribution, more comparisons to alternative pose-injection strategies would strengthen the paper. For example, I recommend adding a baseline that conditions directly on pose parameters (e.g., camera/object pose) rather than using surface-anchored positional embeddings, to better isolate the benefit of SAPE.

2. From the qualitative results, I observe artifacts around the subject–background boundary. In addition, some outputs appear physically implausible (e.g., objects floating above the background). Do the authors have insights into the cause of these failures, and possible ways to mitigate them (e.g., improved harmonization, contact constraints, or stronger background/geometry consistency)?

3. Regarding the data construction pipeline (Fig. 8), the synthesized target images may look unrealistic in some cases, which could introduce bias during training. I suggest using more realistic target images (or adding additional filtering/quality control) to reduce synthetic artifacts and improve real-world generalization.

---

> ### Author Rebuttal · Authors · 2026-03-31
>
> **Q1: Comparisons to Alternative Pose-Injection Strategies**
>
> We appreciate your professional suggestion. Directly using camera/pose parameters as conditioning is indeed an important approach for this task. In **Tab. 1**, the application of Qwen-Edit includes a checkpoint specifically fine-tuned for camera pose control. This allows users to specify camera rotation angles in the prompt, which can be seen as a classical implementation of such methods. Furthermore, in our ablation study baseline, the surface coordinate maps describing the object pose are input as visual tokens rather than SAPE. The comparison with this baseline also illustrates the effectiveness of SAPE.
>
> **Q2: Further Discussion about Generated Images**
>
> We also observed this issue in experiments. The primary cause of physical implausibility in some generated images appears to be data bias. Our data synthesis pipeline was designed to improve background diversity, but it did produce some samples with inappropriate physical interactions. This is due to both the limits of current inpainting models and the fact that many objects in the original images are in rare poses.
>
> We believe the most fundamental solution is to improve data quality, which we describe in our response to **Q3**. Another practical solution is to design a looser pose control method, allowing the model to adaptively adjust objects to similar, more "classical" poses. This will help improve the quality of interactions.
>
> **Q3: Further Discussion about Data Construction**
>
> We appreciate this insightful suggestion. As noted in our response to **Q2**, the current synthetic dataset does exhibit certain limitations in physical realism. Beyond implementing additional filtering as suggested, we believe that leveraging video generation models is another promising direction, particularly for industry developers with more computational resources. This approach could enable a vast range of subject diversity. Nonetheless, as the first public dataset for pose-controlled customization, we believe our dataset can provide a foundational starting point for future researches in this field.

---

> > ### Author Rebuttal · Reviewer_AtXa · 2026-04-01
> >
> > Thanks for the authors' rebuttal; most concerns are well addressed.  I will keep my rating and learn to accept this paper.

---

### Official Review · Reviewer_mXap · 2026-03-13

**Soundness:** 2
**Presentation:** 3
**Significance:** 3
**Originality:** 3
**Overall Recommendation:** 4
**Confidence:** 3

**Summary:**

This paper proposes Pose-ICL, a framework for pose-controllable subject customization in image generation. The key idea is to inject explicit 3D structure into a DiT backbone via Surface-Anchored Position Embedding (SAPE), which encodes surface coordinates from a volumetric bounding box into attention. The method is presented as a tuning-free subject customization approach at inference time and is evaluated on both synthetic 3D assets and real-world subjects, with reported gains in pose accuracy and identity consistency over existing customization, pose-control, and editing-based baselines.

**Compliance With Llm Reviewing Policy:**

Affirmed.

**Key Questions For Authors:**

1. Please clarify more precisely what is meant by “tuning-free.” Does this mean only no subject-specific optimization at test time?
2. How robust is the method to errors in pose estimation and bounding box construction?
3. How much of the gain comes from SAPE itself versus the curated training data and synthetic data pipeline?
4. Can the method generalize to less rigid objects or categories beyond the limited real-world set considered here?
5. Can the authors provide stronger evidence that the learned mechanism goes beyond local correspondence matching and captures more global 3D structure?

**Limitations:**

yes

**Strengths And Weaknesses:**

Strengths
1.The idea of injecting 3D pose information through positional embeddings is interesting and technically clean.
2.The method is well motivated, and the discussion of why 2D-native customization methods fail under pose changes is compelling.
3.The empirical results are strong overall, especially the large gains in pose accuracy relative to pose-control baselines.
4.The ablation study is fairly thorough and helps justify the design choices behind SAPE, shallow injection, and sub-patch encoding.
5.The dataset contribution is potentially useful for future work in this area.
Weaknesses
1.The term “tuning-free” is somewhat misleading, since the framework still requires task-level training with LoRA; it is only tuning-free for new subjects at inference time.
2.The comparison is not fully controlled, since baselines use different backbones, adaptation settings, and inference regimes. Some of the gains may partly reflect these differences rather than only the proposed 3D-aware mechanism.
3.The evidence for true “volumetric understanding” is suggestive but not conclusive. The attention visualizations are useful, but they do not fully establish that the model learns a robust 3D representation rather than a strong pose-conditioned matching prior.
4.Pose accuracy relies on an external pose estimator, and the reliability of this metric on generated images is not fully discussed.
5.The dataset construction pipeline introduces synthetic backgrounds and compositing, which may bias the learned behavior. The paper acknowledges this in the appendix, but this limitation deserves more emphasis in the main text.
6.The real-world evaluation is relatively narrow, with only four real-world categories, which limits the strength of the generalization claims.

---

> ### Author Rebuttal · Authors · 2026-03-31
>
> **Q1: Clarification about "Tuning-Free"**
>
> Thank you for your time and thoughtful assessment. Your understanding is correct: Our framework necessitates task-level training and requires no subject-specific optimization or fine-tuning at test time. We will further emphasize this in the revised manuscript.
>
> **Q2: Robustness to Noisy Pose Annotation**
>
> Please refer to our response to **#R1Q1 (9uwc)**.
>
> **Q3: Clarification about Drivers of Improvement**
>
> We appreciate the reviewer’s insightful question regarding the source of performance gains. To clarify this, we would like to highlight the results from our ablation study. In **Tab. 2**, the baseline is trained on the same curated dataset but processes the surface coordinate maps $S$ as visual input context rather than employing the SAPE mechanism.
>
> Using SceneDesigner as a comparison (which utilizes a dataset of similar scale: 0.12M vs. our 0.15M), the baseline reduces the pose error by 2.82. In contrast, the complete application of SAPE (**Ours**) achieves a larger error reduction of 14.95. These results indicate that while the task-specific dataset certainly provides performance gains, the decisive advantage of our method is established by the SAPE.
>
> **Q4: Further Discussion about Less Rigid Objects**
>
> This study primarily focuses on rigid objects. In non-rigid scenarios, some local components of the object may undergo deformation across different reference images. We use the first object in **Fig. 13** to simulate this scenario, where the robot's arms are positioned in different poses across reference views (**The figure is available at this [link](https://github.com/icml2026sub16016/rebuttal_pic/blob/main/R2Q4.png)**).
>
> The results show that while the overall target pose remains controlled, the specific pose of the arms tends to align with one of the reference images rather than achieving arbitrary deformation. We attribute this to the same reason discussed in our response to **#R1Q1 (9uwc)**. While this shows practical generalization, we admit it is still limited. We believe fully controlling non-rigid objects is an exciting goal, which may need a more flexible control proxy than simple 3D bounding boxes.
>
> **Q5: Further Discussion about "3D-awareness"**
>
> We appreciate this valuable advice on our work. There are cases in our experimental results where the view change is extreme (**The figure is available at this [link](https://github.com/icml2026sub16016/rebuttal_pic/blob/main/R2Q5.png)**). For instance, when a single reference image only shows the back of a cabinet and the target pose requires the front view, there is almost zero local feature overlap. Nevertheless, the model successfully understands the spatial requirement and renders a plausible front view with drawers. We consider these results as evidence of the model’s 3D-awareness. However, pursuing 3D-awareness on a 2D-native foundation model is challenging, and enhancing feature correspondence is the primary way our work achieves it, which is why **Fig. 5** emphasizes this point. We are happy to add these results to the manuscript.
>
> **Response to Additional Points in Weaknesses**
>
> We noticed that several points raised in the Weaknesses were not listed in the Questions. We hope the following responses can address your concerns on these points to some extent.
>
> (**Weakness 2**) It is indeed computationally expensive to retrain all methods under the same backbone. Therefore, we included stronger foundation models like Qwen-Edit in comparison to better demonstrate the performance of Pose-ICL on this task.
>
> (**Weakness 4**) The practice of using Orient Anything to measure pose error is consistent with SceneDesigner. The original paper of Orient Anything also included generated images as part of the test set, and their Section 7.2 discusses its effectiveness on generated images. We are happy to further cite their conclusions in the paper.
>
> (**Weakness 5**) Please refer to our response to **#R3Q2+Q3 (AtXa)**.

---

> > ### Author Rebuttal · Reviewer_mXap · 2026-04-04
> >
> > My major concerns have been well resolved.

---

### Official Review · Reviewer_9uwc · 2026-03-13

**Soundness:** 3
**Presentation:** 3
**Significance:** 3
**Originality:** 2
**Overall Recommendation:** 4
**Confidence:** 2

**Summary:**

This paper proposes Pose-ICL, a tuning-free framework for pose-controllable subject customization. The method introduces a 3D-aware in-context learning paradigm that leverages paired image–pose references to adapt to new subjects. The key component, Surface-Anchored Position Embedding (SAPE) anchors image tokens to volumetric surface coordinates to provide explicit 3D awareness. Experiments on both 3D assets and real world subjects show improvements in pose accuracy and identity consistency.

**Compliance With Llm Reviewing Policy:**

Affirmed.

**Key Questions For Authors:**

How sensitive is the method to noise in pose annotations or imperfect pose estimation?

Can the authors provide experiments on how Pose-ICL performs when the pose distribution differs significantly from the reference poses?

Overall, I find the paper reasonably well presented with solid experimental validation, and I lean toward a weak accept.

**Limitations:**

The approach relies on pose-conditioned references, which may limit applicability when accurate pose annotations are unavailable.

**Strengths And Weaknesses:**

**Strengths**

The paper addresses pose-controllable subject customization, which is an important problem in personalized image generation.

The proposed 3D-aware in-context learning formulation is interesting.

The paper is well written and easy to follow.

The experimental analysis is relatively comprehensive, including ablation studies and evaluations on both synthetic and real subjects.

**Weaknesses**

The novelty mainly lies in integrating 3D-aware embeddings with in-context learning, and the overall framework still relies heavily on existing DiT  architectures.

The method depends on paired image–pose references, and it is unclear how robust it is when pose annotations are noisy or inaccurate.

The approach relies on pose-conditioned references, which may limit applicability when accurate pose annotations are unavailable.

---

> ### Author Rebuttal · Authors · 2026-03-31
>
> **Q1: Robustness to Noisy Pose Annotation**
>
> We appreciate the reviewer's insightful comment. To investigate the model's robustness against pose estimation errors, we conducted additional experiments by injecting varying degrees of noise ($0^\circ, \pm10^\circ, \pm20^\circ, \pm30^\circ$) into the pose annotations of one randomly selected reference image. We report the pose error (**POSE**) and DINO similarity (**DINO-I**) across 10 test objects (sampling 15 images per noise level) to evaluate control accuracy and identity fidelity, respectively.
>
> | Pose Noise | POSE↓ | DINO-I ↑ |
> | :--- | :---: | :---: |
> | 0° | 2.18° | 0.777 |
> | ±10° | 6.73° | 0.767 |
> | ±20° | 11.05° | 0.752 |
> | ±30° | 16.67° | 0.742 |
>
> As can be seen, while increased pose noise naturally leads to higher pose error (**POSE**), the identity fidelity (**DINO-I**) remains relatively stable. This robustness is corroborated by both quantitative metrics and qualitative observations. (**The figure is available at this [link](https://github.com/icml2026sub16016/rebuttal_pic/blob/main/R1Q1.png)**)
>
> We attribute this robustness to the feature interaction mechanism of the attention layers. As formulated in **Eq. 4**, the attention score between two features is jointly modulated by the SAPE and the dot product of query $q$ and key $k$. When two reference features $a'$ and $b'$ provide conflicting information due to noisy pose annotations, the generated feature $c'$ will preferentially attend to the feature with the higher score rather than mechanically blending them. This competitive mechanism ensures that the model maintains structural integrity even under imperfect pose guidance. We will include these results and discussions in the final manuscript.
>
> **Q2: Robustness to Pose Gap between Ref. and Target Images**
>
> This is an important question. The gap between reference and target poses indeed impacts the effectiveness of pose control. In **Appendix C.2**, we have provided a quantitative analysis of this issue, demonstrating that the stability of pose control tends to decrease as the pose difference increases. Furthermore, we show that using multiple reference images can effectively mitigate this limitation.

---

> > ### Author Rebuttal · Reviewer_9uwc · 2026-04-04
> >
> > My concerns have been adequately addressed. Keep my score.

---

### Decision · Program_Chairs · 2026-04-30

**Decision:**

Accept (regular)

**Comment:**

This paper received all positive scores.
Hence, AC recommends accepting this paper.
It is recommended to include all valuable points raised during the rebuttal period.